# Observation of laser-assisted electron scattering in superfluid helium

Leonhard Treiber[1], Bernhard Thaler[1], Pascal Heim[1], Michael Stadlhofer[1], Reika Kanya[2,3], Markus Kitzler-Zeiler[4] & Markus Koch [1]✉

Laser-assisted electron scattering (LAES), a light–matter interaction process that facilitates energy transfer between strong light fields and free electrons, has so far been observed only in gas phase. Here we report on the observation of LAES at condensed phase particle densities, for which we create nano-structured systems consisting of a single atom or molecule surrounded by a superfluid He shell of variable thickness (32–340 Å). We observe that free electrons, generated by femtosecond strong-field ionization of the core particle, can gain several tens of photon energies due to multiple LAES processes within the liquid He shell. Supported by Monte Carlo 3D LAES and elastic scattering simulations, these results provide the first insight into the interplay of LAES energy gain/loss and dissipative electron movement in a liquid. Condensed-phase LAES creates new possibilities for space-time studies of solids and for real-time tracing of free electrons in liquids.

[1] Graz University of Technology, Institute of Experimental Physics, Graz, Austria. [2] Department of Chemistry, Faculty of Science, Tokyo Metropolitan University, Hachioji-shi, Tokyo, Japan. [3] JST PRESTO, Hachioji-shi, Tokyo, Japan. [4] Technische Universität Wien, Photonics Institute, Vienna, Austria. ✉email: markus.koch@tugraz.at

The investigation of atomic-scale processes with high spatio-temporal resolution is key to the understanding and development of materials. While pulsed light sources have been developed to provide attosecond temporal resolution[1], the diffraction limit of light waves prohibits the improvement of the spatial resolution below the ten-nanometer range. Electron probes, in contrast, allow for subatomic spatial resolution due to their picometer deBroglie wavelength, and can achieve high temporal resolution[2–9]. Time-domain shaping of electron pulses is based on the transfer of energy between electromagnetic radiation and free electrons, which is manifested in various phenomena, such as bremsstrahlung, Smith-Purcell radiation[10,11], Cerenkov radiation[12], or Compton scattering[13,14]. Electron–photon coupling is furthermore key to the development of novel light sources like free electron lasers[15] or high–harmonic generation[16], and to ultrafast structural probing like high–harmonic spectroscopy[17] or laser-induced electron diffraction[18]. While few- and sub-femtosecond electron pulses[19,20] and pulse trains[21,22] could be generated through light-field manipulation, the time resolution achievable with these electron pulses suffers from velocity dispersion and Coulomb repulsion[20].

LAES is a light–matter interaction process that offers a unique advantage for time-resolved electron probes by combining time-domain shaping of electron pulses with structural probing. In LAES, free electrons that scatter off neutral atoms or molecules in the presence of a strong laser field, can increase (inverse bremsstrahlung) or decrease (stimulated bremsstrahlung) their kinetic energy by multiples of the photon energy ($\pm n\hbar\omega$)[23–25]. Structural information of the scattering object is encoded in the angular distribution of the accelerated/decelerated electrons[23,26]. Importantly, the energy modulation only takes place during the time window in which the short laser pulse overlaps with the much longer electron pulse within the sample. LAES can thus be viewed as an optical gating technique that allows to record scattering-snapshots at precisely defined times. The capability of LAES to analyze structural dynamics with subparticle spatial resolution (~1 pm) at the timescale of electron dynamics (<10 fs) was recently demonstrated in the gas phase[23,26]. Other strong-field phenomena like high-order harmonic generation[27] have been extended from the gas phase to solid-state systems, providing insight into the attosecond electron dynamics and non-equilibrium situations in band structures. Also, the laser-assisted photoelectric effect was demonstrated from the surface of a solid[28], allowing to map the electron emission process with attosecond resolution[29]. LAES, in contrast, where an electron probes the structure of neutrals far away from its origin, has evaded observation in the condensed phase so far, so that its potential for advancing time-resolved structural probing at high particle densities remains unexplored.

Here, we demonstrate that LAES can be observed at condensed-phase particle densities of $2 \cdot 10^{22}$ cm$^{-3}$, for which we create core–shell nanostructures, consisting of a single atom/molecule located inside a superfluid He droplet (He$_N$)[30,31]. This system provides three unique advantages: First, the droplet size and thus the LAES interaction shell thickness underlies a well defined distribution, the mean of which can be varied with Angstrom resolution[30]. Second, the high strong-field ionization threshold of He[32] enables high laser intensities to increase the LAES probability without solvent ionization. Third, energy dissipation of electrons propagating inside He$_N$ is very low[33]. Such advantages recently enabled the application of above threshold ionization (ATI) to He droplets[34]. We have chosen experimental conditions to work in the multiple scattering regime in order to characterize the interplay of LAES acceleration/deceleration and dissipative electron movement within the He shell; as a consequence, our experiment does not provide information about the electron angular distribution.

## Results

**Observation of LAES within a He droplet.** To measure the energy gain of electrons through LAES within the liquid He shell of our core-shell system, we perform strong-field photoionization with femtosecond laser pulses and compare two photoelectron spectra that are recorded under the same laser pulse conditions: First, the ATI spectrum of a bare, gas-phase atom/molecule and, second, the LAES spectrum obtained with the same atom/molecule embedded inside a He$_N$. The He droplets, which have a radius of a few nanometer, are created by supersonic expansion of He gas through a cryogenic nozzle and are loaded with single dopant atoms or molecules through the pickup technique[30,31], as described in the Methods section below. Figure 1a–c shows the two types of spectra for three species: Indium (In) atoms, xenon (Xe) atoms, and acetone (AC) molecules. For all three species, the LAES spectrum shows significantly higher electron energies than the ATI spectrum, and both types of spectra—ATI and LAES—

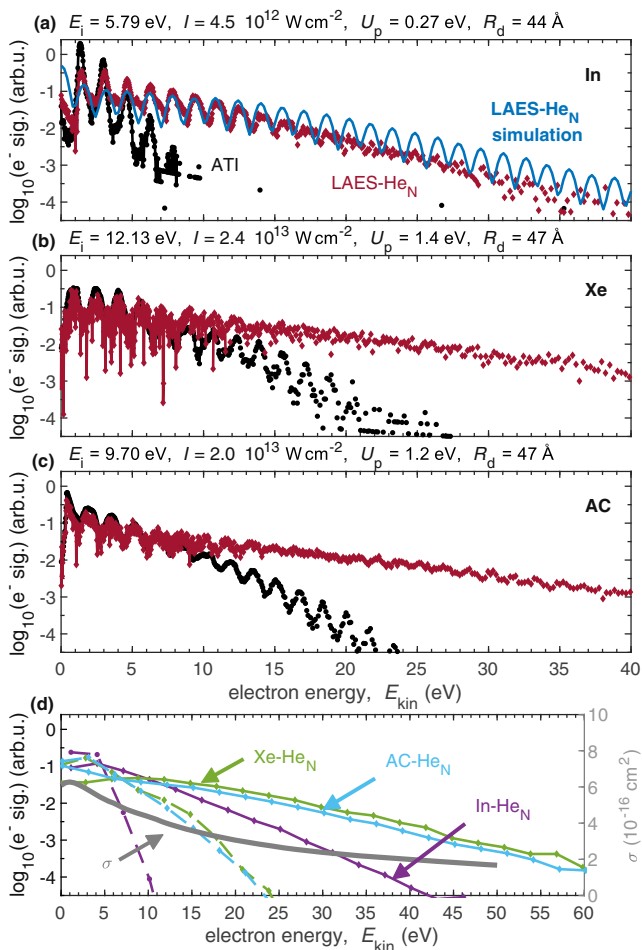

**Fig. 1 Comparison of experimental electron spectra obtained by strong-field ionization with 800 nm light (1.55 eV photon energy) of different species in gas phase (ATI spectra, black) and inside He$_N$ (LAES spectra, red).** (a) In atoms, (b) Xe atoms, (c) acetone (AC) molecules. The spectra are area-normalized in order to account for the reduced ionization energy inside a He$_N$[33]. Above each plot the values of the ionization energy, $E_i$, laser intensity, $I$, ponderomotive potential, $U_P$, and droplet radius, $R_d$, are listed. Panel (a) additionally shows a spectrum obtained by a Monte Carlo 3D LAES simulation. (LAES simulation, blue, area normalized). **d** ATI spectra (dashed lines) and LAES spectra (solid lines) as in a-c but with 3 eV binning (left ordinate), and cross section for total elastic electron scattering of electrons and He (gray line, right ordinate)[38].

show equidistant signal modulations with a peak distance of 1.55 eV, corresponding to the central laser wavelength of 800 nm (1.55 eV photon energy). Closer inspection of the area-normalized spectra shows that, in addition to the higher energies of the LAES spectrum, the ATI signal exceeds the LAES signal at low energies up to ~5 eV, indicating a shift of the electron energy distribution towards higher kinetic energies due to the presence of the He shell.

In order to identify the process causing this strong electron acceleration, we simulate the interaction of the In-He$_N$ system with a light pulse under the same conditions as in the experiment. As described in detail in the Methods section, we obtain the initial electron kinetic energy distribution by assuming tunnel ionization by the laser field[35] and simulate subsequent binary LAES events with He atoms by applying the Kroll-Watson theory[36]. Our simulation calculates 3D electron trajectories within a He droplet of radius $R_d$ and applies a Monte Carlo approach for the LAES events. The simulated LAES spectrum (Fig. 1a, blue curve) shows the kinetic energy distribution of electrons within 400 fs, the time-window of the simulation. The good agreement of the simulated spectrum with the experiment strongly indicates that the observed electron acceleration is due to LAES.

In order to investigate the influence of the dopant species that serves as electron source through strong-field ionization, we compare the In, Xe and AC spectra (Fig. 1a–c). We use a higher laser intensity $I$ for Xe and AC due to the higher ionization energy $E_i$, as compared to In, which is reflected by ATI spectra that extend to higher energies. Smoothed LAES and ATI spectra are compared in Fig. 1d. The similarity of the Xe and AC spectra, for which a very similar laser intensity was used, indicates that the species from which the electrons originate has very little influence. Instead, the LAES energy gain is larger for Xe and AC (e.g., at a signal level of $10^{-3}$: 25–30 eV gain), compared to In (20 eV gain), because of the higher laser intensities used for Xe and AC. These observations indicate that the laser intensity dictates the energy gain.

In addition to the LAES energy gain, insight into the dissipative electron movement within the liquid He shell can be obtained from the equidistant peak structure of the LAES spectra (Fig. 1a–c). Kinetic energy of an electron can be dissipated to the He droplet through binary collisions with He atoms and through a collective excitation of the droplet. While elastic collision with a He atom reduces the electron kinetic energy by ~0.06% due to energy and momentum conservation, collective He$_N$ excitations carry < 2 meV energy[31]. The pronounced contrast of the LAES peaks in Fig. 1 thus demonstrates that energy dissipation plays a subordinate role compared to LAES energy gain for the relatively small droplets ($R_d \approx 45$ Å radius) used in these measurements. Furthermore, the absence of a kink in the yield at or above 20 eV, the energy threshold of electronic He excitations[37], shows that inelastic interactions are insignificant, which is in agreement with the much lower cross section for inelastic as compared to elastic interaction[38].

**Droplet size effects.** We now investigate the influence of the He shell thickness on the LAES spectrum in order to deepen our insight into the interplay of light-induced energy gain/loss and dissipative energy losses. The He droplet approach allows to change the He shell thickness around the atom/molecule to be ionized by varying the droplet source temperature. Since LAES processes require electron–He scattering in the presence of laser light, information about the electron transit time through the He shell can be gained from the droplet-size dependence of the LAES spectra. Figure 2a shows LAES spectra obtained with In atoms inside He droplets with radii from $R_d = 32$ Å to $R_d = 340$ Å. The

energy gain continuously increases with the He shell thickness for the accessible range of droplet radii. The maximum kinetic energy doubles from 50 eV for the smallest droplets to 100 eV for the largest ones, compared to a maximum energy of the ATI spectrum of about 15 eV. This continuous increase provides a first indication that the transit time distribution, which is a result of stochastic electron trajectories, is comparable to the laser pulse duration, at least up to $R_d \approx 76$ Å.

While the LAES energy gain is restricted to the time window of the laser pulse, the electron dissipates energy as long as it propagates within the He droplet. Since the energy transfer in single collisions with He atoms is low, energy dissipation influences the modulation contrast of the LAES signal. A close-up of the LAES spectra in the low-energy region in Fig. 2b allows to evaluate the dependence of this contrast on the droplet size. We find that the contrast decreases steadily from the smallest droplets, where it equals the contrast of the gas-phase ATI peaks, until it vanishes completely for the largest droplets. We ascribe this blurring to energy dissipation of the electron within the He shell, which has increasing influence on the spectra for larger droplets. Despite the energy dissipation, the thickest He shell

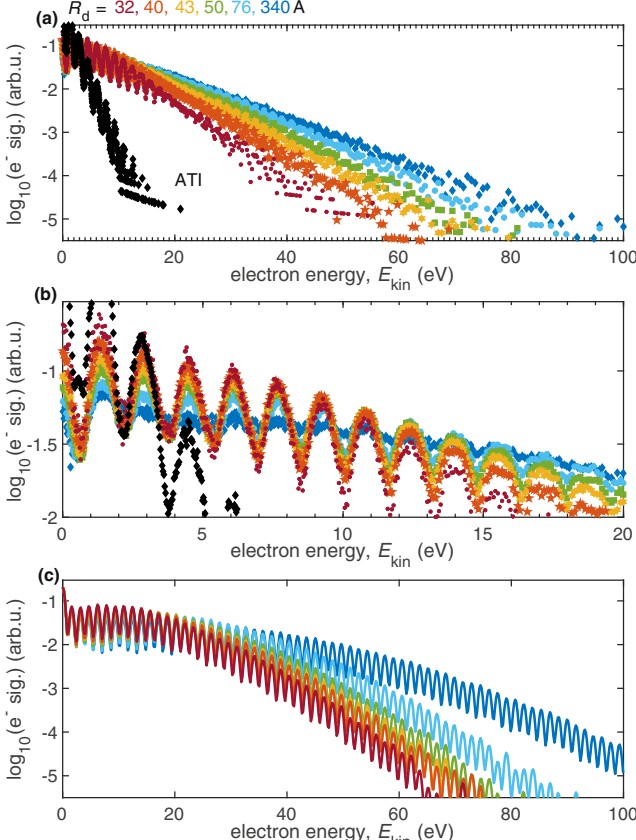

**Fig. 2 Dependence of LAES spectra on the droplet size, for droplet radii between $R_d = 32$ Å and $R_d = 340$ Å ($R_d$ values are calculated from the mean values of the droplet size distributions[30]). a** The experimental spectra are obtained with In atoms at $I = 1.1 \cdot 10^{13}$ Wcm$^{-2}$ and show a pronounced increase of the LAES energy gain with He shell thickness. In addition, the gas-phase ATI spectrum is shown for comparison. The abrupt increase of the droplet radius to $R_d = 340$ Å for the lowest droplet source temperature ($T_0 = 10$ K) is due to the changing character of the supersonic expansion from sub- to supercritical in this temperature regime[30]. **b** Close-up of the low-energy region of (**a**). **c** Simulated LAES spectra for different droplet sizes, under the same conditions as in (**a**). The spectra are area normalized.

($R_d$ = 340 Å) supports the highest LAES energy gain, emphasizing the dominance of the light-driven electron energy modulation over dissipative energy loss.

Simulated LAES spectra for different droplet sizes, shown in Fig. 2c, also reveal a very pronounced droplet-size dependence of the electron spectrum. As in the experiment, the energy gain continuously increases with droplet size because larger droplets allow for an increased number of LAES events within the duration of the laser pulse. For comparison of the experiment to the simulations it is important to realize that the droplet sizes specified for the experiment (Fig. 2a) are subject to uncertainty because (i) the generation process of the droplets results in a log-normal size distribution and (ii) the probability to load a droplet with an atom/molecule follows a Poissonian distribution[30]. The fact that the experiment shows a slightly lower energy gain maximum for $R_d$ = 340 Å (Fig. 2a), as compared to the simulations (Fig. 2c), indicates that the mean droplet radius of the ensemble observed in the experiment is slightly smaller than 340 Å. Additional deviations might arise from the assumption of the simulations that the dopant is located at the center of the droplet, whereas the experiment might average over a spatial dopant distribution given by a flat holding potential[39]. Apart from this minor difference, Figure 2 reveals very good agreement of the predicted and observed droplet size dependence of the LAES process.

**Characterization of the electron–helium interaction.** For further insight into the electron propagation through the He shell we retrieve characteristic parameters from our LAES simulation. In addition, we perform simple 3D elastic scattering simulations without considering the light field for electron trajectories much longer than the 400 fs used in the LAES simulations. In addition, to obtain information about the total number of elastic scattering events and the corresponding energy distribution (for details see the Methods section). Figure 3a shows the ratio of ejected electrons over time for different droplet sizes. It can be seen that the ratio of ejected electrons within the laser pulse duration (gray line in Fig. 3a) depends strongly on the droplet size. The median value for the electron transit time through the liquid He layer, corresponding to an electron ejection ratio of 0.5, increases from 11 fs for $R_d$ = 32 Å, to 20 fs for $R_d$ = 76 Å, and to 164 fs for $R_d$ = 340 Å. The simulated ratios of ejected electrons level off for the smaller droplets at ~85%, indicating that ~15% of the electrons have not left the droplet by the end of the simulated time window, although this value might be subject to uncertainty due to incomplete literature values for the differential scattering cross sections of very slow electrons[40].

The probability distributions of laser-assisted scattering events (Fig. 3b) give further insight into the droplet size dependency. The mean number of scattering events increases by a factor of 4 from 6 for $R_d$ = 32 Å to 24 for $R_d$ = 340 Å.

Finally, we look into the dissipative electron movement and therefore consider purely elastic scattering of 5 eV electrons and the scattering event distribution after ejection from the droplet (long interaction times beyond 400 fs, Fig. 3c, d). Comparing these distributions to the mean number of LAES events within the pulse duration in Fig. 3b, it is obvious, that for the largest droplets, the majority of scattering events happen after the laser pulse.

## Discussion
Comparison of strong-field ionization spectra of atoms/molecules in gas phase and inside He droplets reveals that the presence of a nanometer-thick layer of superfluid He around the ionized particle leads to a significant increase of the electron kinetic energies.

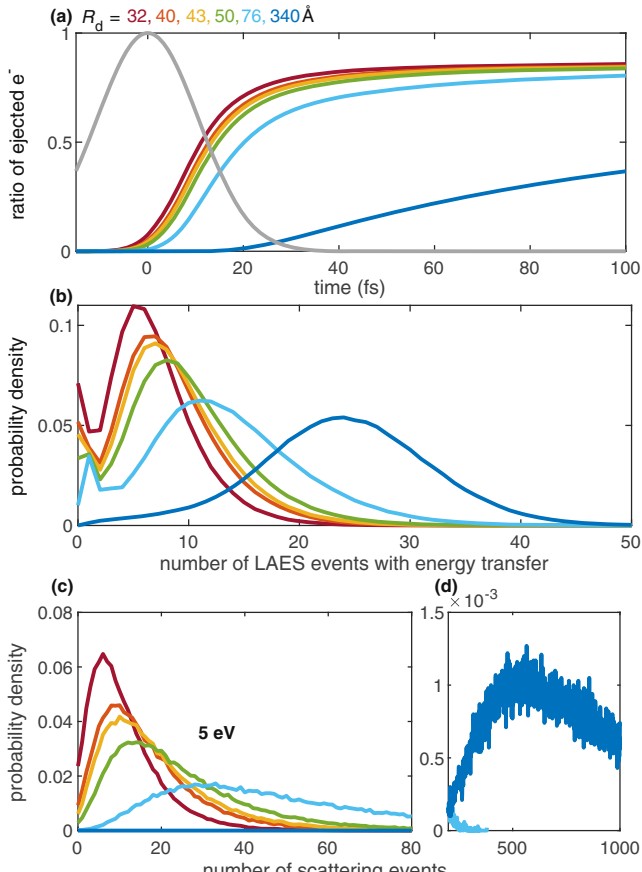

**Fig. 3 LAES simulation of electron trajectories inside a He droplet for different droplet radii $R_d$. a** Ratio of ejected electrons (e⁻) as function of time for different $R_d$. For comparison, a Gaussian laser pulse envelope is shown in gray. **b** Number of laser-assisted scattering events with energy transfer as function of the droplet radius. **c, d** Probability distribution of total elastic scattering events, i.e., without laser field and without temporal limit, for $E_{kin}$ = 5 eV.

The following observations, in combination with Monte Carlo 3D LAES simulations, lead us to the conclusion that the electron acceleration is due to multiple LAES processes within the He layer: (i) The simulated electron spectrum for strong-field ionization of the In-He$_N$ system agrees very well with the observed spectrum in terms of slope and equidistant peak structure (Fig. 1a), identifying LAES as the process responsible for electron acceleration. (ii) The energy gain strongly increases with droplet size (Fig. 2). This behavior observed for strong-field ionization is in contrast to weak-field ionization inside He droplets, where the photoelectron spectrum is either droplet-size independent because it is influenced only by the structure of the immediate environment of the dopant, the solvation shell[41], or develops a low-energy band revealing significant energy loss of electrons in larger droplets[42]. In the current situation, the energy gain of the electron is related to the number of light-mediated binary electron–He-atom collisions at a distance from the remaining ion, which increases with growing droplet size. (iii) Comparison of three different species shows that the laser intensity has the strongest influence on the LAES energy gain, while the ionization energy plays a negligible role (Fig. 1). This can be explained by an increased LAES probability due to increased photon flux. (iv) Our simulations predict on average between 6 and 24 sequential LAES processes for the combination of droplet sizes and laser pulse parameters used in the experiments.

A crucial factor for the observation of LAES in the condensed phase is the interplay of LAES energy gain/loss and dissipative energy loss as a function of the thickness of the material. In the experiment we observe that the LAES energy gain increases continuously over the whole range of investigated droplet sizes ($R_d = 32$ Å to $R_d = 340$ Å, Fig. 2). The LAES simulations agree well with this droplet-size dependent increase and predict correspondingly a rise of the median transit-time from 11 fs ($R_d = 32$ Å) to 164 fs ($R_d = 340$ Å). The LAES interaction time is thus determined by the droplet size in small droplets and by the laser pulse duration in large droplets (Fig. 3).

Finally, we want to focus on the dissipative electron movement. Considering purely elastic scattering (Fig. 3c, d), on average, 5 eV electrons undergo 10 collisions inside the smallest droplets ($R_d = 32$ Å), resulting in an energy loss of 30 meV (0.06% energy loss per collision). Inside the largest droplets ($R_d = 340$ Å) they loose, on average, 2 eV after 830 elastic collisions. Comparing these values to the 1.55 eV distance of LAES peaks, the signal contrast is expected to be the same as that of the gas-phase ATI spectrum for the smallest droplets, while it can be expected to fully smear out for the largest droplets, in agreement with our measurements in Fig. 2b. However, the simulated electron energy loss of 130 meV (45 collisions) for $R_d = 76$ Å, seems insufficient to explain the observed ~50% contrast reduction (around $E_{kin} = 5$ eV) in Fig. 2b. This discrepancy points towards shortcomings of the simulation that are currently neglected: Excitation of collective droplet modes[30,31], transit-time increase due to Coulomb interaction between the ion core and the electron, or additional blurring of the LAES peaks due to sequential energy-gain–energy-loss processes induced by the femtosecond laser pulse with the bandwidth of 125 meV. Nevertheless, the most important observation is that the largest He droplet (thickest He layer, $R_d = 340$ Å) yields the fastest electrons, proving that energy gain through multiple LAES processes effectively dominates over energy dissipation for propagation distances of several tens of nanometers.

In conclusion, we have demonstrated that LAES can be observed with femtosecond laser pulses in the condensed phase at particle densities of $2 \cdot 10^{22}$ cm$^{-3}$. We show that electrons can be accelerated to high kinetic energies through multiple LAES processes and support our interpretations with Monte Carlo 3D LAES simulations. Our results indicate that LAES is a strong-field light–matter interaction process that is, in analogy to high harmonic generation[27], capable of spatio-temporal analysis of solids. It can be anticipated that LAES has the potential to significantly increase the temporal resolution of electron probes through optical gating, thereby merging temporal selection via velocity modulation of electrons with ultrashort laser pulses (as demonstrated here), and structural analysis that can be extracted from the electron angular distributions[23,26].

The significant acceleration of electrons and its dominance over energy dissipation within liquid He is likely related to the outstanding properties of this rare-gas element: The application of high light-field intensities resulting in strong LAES energy gain is enabled by the exceptionally high ionization energy of He and the high excitation energy prevents inelastic electron collisions up to 20 eV. In heavier rare-gas clusters, LAES can be expected, too, albeit less pronounced. The contribution of the droplets' superfluid character to the observed energy modulation cannot be deduced from the present results and remains to be investigated, for example with non-superfluid $^3$He droplets or mixed $^3$He/$^4$He droplets[43]. It will also be important to investigate the ratio of light-induced energy gain and energy dissipation in other materials, like molecular, metal or semiconductor clusters, the creation of which is facilitated by the very flexible opportunities provided by the He droplet approach for the creation of tailor-made

bi-material core-shell nanostructures within the droplet[44,45]. Photoionization of the core will allow to observe LAES-acceleration and energy dissipation within the shell material. Furthermore, extension to a pump-probe configuration with few-cycle pulses (~5 fs duration) should enable tracing of electron propagation within the target material.

## Methods

**Helium nanodroplet generation and particle pickup**. We generate superfluid helium nanodroplets (He$_N$) in a supersonic expansion of high-purity He gas through a cooled nozzle (5 μm diameter, 40 bar stagnation pressure) into vacuum. Variation of the nozzle temperature between 10 and 20 K allows us to change the mean droplet size in the range of $\bar{N} = 3.0 \cdot 10^3 - 3.7 \cdot 10^6$ He atoms per droplet[30], corresponding to a droplet radius of $R_d = 32 - 340$ Å. After formation, evaporative cooling results in superfluid droplets at a temperature of about 0.4 K. We load the droplets with single dopant atoms or molecules by passing them through a resistively heated dopant oven (In), or a gas pickup cell (Xe, acetone). We further monitor the pickup conditions by recording the monomer, dimer, and trimer ion signals (e.g., In$^+$, In$_2^+$, In$_3^+$) with a quadrupole mass spectrometer as a function of the current of the resistively heated pickup cell. When changing the droplet size we ensure constant pickup conditions by adapting the particle density within the pickup region accordingly. Since loading the He droplets is a statistical process, we have carefully checked if the presence of multiple dopants within one droplet influences the LAES spectra. In the range of, on average, one to three In atoms per droplet we find no significant change of the spectra which can be rationalized by the following two aspects: First, multimer formation due to van der Waals interaction between individual dopants leads to single ionization centers even in multiply doped droplets and, second, the initial photoelectron spectrum of these multimers is size-independent and similar to that of the monomer.

**Strong-field photoionization and detection of LAES spectra**. We ionize the guest atom/molecule inside a droplet with femtosecond laser pulses from an amplified Ti:sapphire laser system (800 nm center wavelength, 25 fs pulse duration, 3 kHz repetition rate, 1 mJ maximum pulse energy), which we focus to obtain intensities of $I \leq 3 \cdot 10^{13}$ Wcm$^{-2}$, as indicated on top of Fig. 1a–c. The pulse duration is measured with a single-shot autocorrelator and the intensity is calibrated using the $U_P$ energy shift of electrons generated by ATI of H$_2$O at a pressure of $1 \cdot 10^{-7}$ mbar[46]. Laser-ionization of the doped droplets takes place inside the extraction region of a magnetic-bottle time-of-flight spectrometer and electron spectra are computed from flight-time measurements[33,39]. We compare LAES spectra of atoms/molecules inside the droplets to ATI spectra of bare atoms/molecules, which we obtain as effusive beam from the pickup cell by blocking the He droplets. The measurement chamber is operated at a base pressure of $10^{-10}$ mbar.

**Monte Carlo 3D LAES simulations**. In the Monte Carlo 3D LAES simulations, $10^7$ electron trajectories are calculated from $-50$ fs to 400 fs with time steps of 15 as, where time zero is defined to be at the peak of the laser pulse envelope with the FWHM duration of 25 fs. At each time step, LAES probability is evaluated, and energies and directions of scattered electrons are determined on the basis of Kroll-Watson theory[36], with field-free elastic scattering cross sections and corresponding differential cross sections taken from ref. 40. The birth time and the initial canonical momentum of photoelectrons are evaluated by the ADK-type tunnel ionization[35] theory applied to In. Spherical He droplets with a uniform number density of $n = 2.18 \cdot 10^{22}$ cm$^{-3}$ are assumed[47]. The position of the dopant atom/molecule is located at the center of the droplet, and the Coulomb potential from the dopant ion after the tunnel ionization is neglected. The laser intensity distribution within the focal volume is considered whereas neither the droplet-size distribution nor inelastic scattering processes are included. After the trajectory calculations until 400 fs, kinetic energy distributions of the photoelectrons ejected from the droplet are evaluated, the electron spectra are obtained through the convolution by a Gaussian function with a FWHM width of 0.8 eV.

**Monte Carlo 3D elastic scattering simulations (without light field)**. For the 3D scattering simulations, we assume an ensemble of electrons with a fixed kinetic energy $E_{kin}$. The ensemble with an isotropic distribution of initial directions propagates from the droplet center and scatters elastically until it finally exits the droplet. We assume binary electron-He collisions of mono-energetic electrons and neglect acceleration/deceleration due to LAES, as well as momentum transfer in elastic scattering events and inelastic interactions. The propagation distance before a scattering event, $s$ is chosen from the exponential distribution $N(x) = N_0 \cdot e^{-n\sigma x}$ (Lambert-Beer law) as $s = -\frac{\log(R)}{(n\sigma)}$, with $R$ uniformly distributed within the interval $[0, 1]$. Values for the elastic scattering cross section $\sigma$ and angular distribution $\frac{d\sigma}{d\Omega}$ are taken from ref. 48 for electron energies up to 10 eV and from ref. 40 for faster electrons. A constant He density of $n = 2.18 \cdot 10^{22}$ cm$^{-3}$ (ref. 47) is assumed.

## Data availabilty

The electron spectra generated in this study are available in Zenodo with the identifier https://doi.org/10.5281/zenodo.4955228.

## Code availability

The code for simulating the LAES spectra is available from the corresponding author on reasonable request.

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

## Acknowledgements

We acknowledge financial support by the Austrian Science Fund (FWF) under Grants P 33166 and P 28475, as well as support from NAWI Graz. This work was in part supported by JST, PRESTO Grant Number JPMJPR2007, Japan.

## Author contributions

M.K. conceived and designed the experiment with contributions of L.T. and M.K.-Z.; P.H. built the experimental setup with contributions of B.T. and M.K.; L.T., B.T., and M.S. performed the experiment; R.K. performed the Monte Carlo 3D LAES simulations; L.T. performed the Monte Carlo 3D elastic scattering simulations with contributions of P.H.; L.T., B.T., M.K., and M.K.-Z analyzed the data; all authors contributed to the interpretation of the results; L.T. and M.K. wrote the paper.

## Competing interests

The authors declare no competing interests.
