## [Peer Review File · Nature Communications]

Reviewers' Comments:

Reviewer #1:

Remarks to the Author:

This manuscript describes the measurement of the LAES phenomenon in doped He nanodroplets. LAES is a so-far little-used phenomenon in the field of strong-field physics. It describes the exchange of energy between photons and nearly free electrons in the vicinity of neutral molecules (atoms). In 2014, Morimoto, Kanya, and Yamouchi wrote an article about this phenomenon in JCP. They first shot electrons of 1 keV at a molecular sample and measured the diffraction pattern. Using the effect of LAES, the energy of the electrons in the molecular sample was modulated. Thereby, the electrons with the energy modulation of one photon energy ($h\nu$) showed a diffraction pattern that corresponds to the diffraction pattern of the undisturbed molecules.

The authors of the present manuscript show that for doped He nanodroplets there is also an effect of energy modulation. However, the detected electrons are not generated by an external and controlled source but are born internally in the droplet. The effect of ATI (above-threshold-ionisation) is used for this purpose. The doping inside the droplet has a lower ionisation potential than the He, so it can be assumed that the electrons are generated from the impurity.

A comparison of the ATI spectra of the pure In/molecule with the spectra of the doped droplets in Fig. 1 clearly shows a shift to higher kinetic energies. However, it is not clear what the ratios of the spectra were before normalisation to the area and how it was ensured that noise in the detector was not scaled up here. As the size of the droplets increases, the spectrum is also shifted to higher kinetic energies (Fig. 2). The authors explain this qualitatively and quantitatively with simulations.

The simulations show the number of collisions for variable droplet size and variable electron energy. Here it is noticeable that in the time interval of the pulse (25 fs) up to 40 collisions take place.

The experimental results are very good, the simulations add value to these results.

Regarding major claims and novelty: The major claim in the abstract, introduction, and conclusion is that ultrafast electron microscopy (UEM) with LAES gating is, due to this paper, also possible in condensed samples. I find this statement confusing.

Good spatial resolution in UEM requires high kinetic energy of the incoming electrons and high angular information content of the outgoing electrons. Morimoto, Kanya and Yamouchi used 1 keV electrons from an external source and modulated those using LAES, and concentrated on the first sideband for retrieving angular and thus structural information. One could extend the simulations given in Fig. 3 to 1 keV and test the cross-sections, and it will most likely work, but this is not the topic of the paper.

Also, experiments with high energy electrons in condensed media are already done with a high spatial resolution (Ischenko et al. Chem. Rev. 217, 16, 11066 give a good review), so the question to ask is if under those conditions the combination with LAES would work.

The claim that could be made in this manuscript is that the LAES energy-modulation mechanism is observed for the first time in a condensed, superfluid medium and that multiple LAES interactions can take place in He-nanodroplets and high kinetic energy can be achieved by this. This however works in the multiple-scattering regime. That is an important discovery in strong-field physics and could possibly be of interest to a wider community.

Reviewer #2:

Remarks to the Author:

This work reports the experimental observation of enhanced emission of energetic electrons generated by ATI of dopants embedded in helium nanodroplets. Similar electron spectra have recently been reported for pure and doped helium nanodroplets [Phys. Rev. Lett. 125, 093202, arXiv:2102.08150], and should be mentioned. The dependence of the electron spectra on the helium droplet size seems to indicate a crucial role of the helium shell around the dopant in enhancing electron emission. Classical elastic electron-helium scattering simulations are performed to get some insight into the interaction of emitted electrons with the helium nanodroplet. The authors put their results into the perspective of ultrafast microscopy. While the reported experimental findings are very interesting and definitely

deserve to be published, the present discussion suffers from severe shortcomings which have to be remedied before this manuscript should be further be considered for publication in Nature Communications.

1) The given interpretation of the experimental data in terms of LAES is based on a classical simulation of electron-helium binary scattering, which informs about the number of elastic collisions as a function of the initial electron energy and the droplet size. However, since the interaction with the laser pulse is not considered, the connection to LAES remains unclear. As long as the simulation does not provide any information about the enhancement of the emission of energetic electrons by the coupling to the laser field, the interpretation in terms of LAES has to be considered as qualitative / speculative. In particular, it remains unclear why energy absorption by the electrons from the light field dominates over emission of photons leading to electron-energy loss.

2) The validity of the scattering model is very limited; besides the mentioned neglect of the interaction of the electron with its parent ion and with the light field, inelastic scattering is disregarded, although cross sections are about 0.1 \AA^2 At. Data Nucl. Data Tables 94, 603 (2008)]. inelastic scattering of a photoelectron upon a neighboring helium was observed [J. Chem. Phys. 150, 044304 (2019)]. More importantly, it is well known that electron-helium collisions at low energies $\sim 1 \text{ eV}$ are crucially influenced by the collective response of the helium superfluid causing the electron to be trapped in a bubble [Phys. Rev. Lett. 75, 4079]; attachment of electrons to helium droplets at energies as high as 10 eV was observed [J. Chem. Phys. 108, 9327 (1998)]. Electrons were found to recombine with impact-excited He atoms and molecules [J. Phys. Chem. Lett. 2014, 5, 2444, J. Phys. Chem. A 2014, 118, 6642–6647]. Moreover, the assumption made by the authors that dopants are located at the center of the droplets may not be justified, at least for the larger droplet they used. The binding potential of the dopants to the droplets is known to be rather flat which leads to some delocalization of the dopants [Mol. Phys, 1999, 97, 645± 666].

3) The role of multiple dopants being present in one droplet needs to be carefully discussed. Due to the broad size distribution of the He droplets, even at moderate average droplet sizes there are contributions to the signals from much larger droplets which have large pickup cross sections. The authors should specify how they made sure that the droplets were doped by single atoms / molecules at all shown droplet sizes.

4) An important motivation for the present work given by the authors is the relevance of their findings for ultrafast electron microscopy. As they don't show any angle-resolved data, this claim remains unfounded and questionable.

Minor issues:

p. 2: Sentence starting with "Time-domain shaping of electron pulses..."; citing references from 1923 and 1953 in this context seems awkward.

p. 2: Sentence starting with "Consequently, spatial separation of pulse shaping and structural probing..." is unclear; what is meant by "spatial separation of pulse shaping...?"

p. 10: Sentence ending with "...because it is influenced only by the structure of the immediate environment of the dopant, the solvation shell." contradicts previous findings of droplet-size dependent electron spectra [J. Phys. Chem. A, 112, 2008, 9359].

Reviewer #3:

Remarks to the Author:

The work presented by Treiber et al. reports on strong field ionization of atoms and molecules surrounded by a helium layer. The interaction of the generated electrons with the helium in the presence of the strong light field leads to the generation of sidebands in the photoelectron spectrum separated by the photon energy. The authors refer to this process as Laser-assisted electron

scattering (LAES) and claim in the abstract that this has only be observed in the gas phase. However, the process is not fundamentally different from the Laser-Assisted Photoelectric Effect (LAPE), where usually a weak UV laser ionizes species in the presence of a strong laser pulse. These types of experiments have been performed in gas, liquid and solid phase. The experiments as described the authors are thus not the first of this kind in the liquid phase as they claim. Their study does provide some additional insight into the process, i.e. to what extend the size of the helium contributes to the shift of photoelectron energy. In my view, these results are however not of broad interest to warrant publication in Nature Communication.

In general, the authors would benefit from a more exhaustive study of the literature. As a motivation for their work they refer to ultrafast electron microscopy, stating that the best resolution obtained is 60 fs. However, Baum and coworkers in Munich recently recorded electron images and diffraction patterns with attosecond resolution. It remains questionable if the authors will be able to achieve such resolution based on LAES, even more so since they do not provide any indication how they would perform such experiments.

As for the reported experiments, there is one issue the authors should address. They mention that in their simulations on the electron motion thorough helium they neglect the acceleration/deceleration due to LAES, and the momentum transfer. However, they do not mention anything whether the charge of the remaining ion is taken into account and what effect this might have on the calculations.

Reply to Reviewer 1

We thank Reviewer 1 for the positive report, in particular his/her comment on the quality of our experimental results. We much appreciate his/her opinion on the importance of our work for strong-field physics and on the potential interest to a wider community.

Reviewer: *This manuscript describes the measurement of the LAES phenomenon in doped He nanodroplets. LAES is a so-far little-used phenomenon in the field of strong-field physics. It describes the exchange of energy between photons and nearly free electrons in the vicinity of neutral molecules (atoms). In 2014, Morimoto, Kanya, and Yamouchi wrote an article about this phenomenon in JCP. They first shot electrons of 1 keV at a molecular sample and measured the diffraction pattern. Using the effect of LAES, the energy of the electrons in the molecular sample was modulated. Thereby, the electrons with the energy modulation of one photon energy ($h\nu$) showed a diffraction pattern that corresponds to the diffraction pattern of the undisturbed molecules.*

The authors of the present manuscript show that for doped He nanodroplets there is also an effect of energy modulation. However, the detected electrons are not generated by an external and controlled source but are born internally in the droplet. The effect of ATI (above-threshold-ionisation) is used for this purpose. The doping inside the droplet has a lower ionisation potential than the He, so it can be assumed that the electrons are generated from the impurity.

A comparison of the ATI spectra of the pure In/molecule with the spectra of the doped droplets in Fig. 1 clearly shows a shift to higher kinetic energies. However, it is not clear what the ratios of the spectra were before normalisation to the area and how it was ensured that noise in the detector was not scaled up here.

Response: The spectra without normalization are shown in Figure 1. The difference to the area-normalized spectra (Figure 1 in the manuscript) is essentially a weaker LAES signal for Xe and AC (Figure 1b, c), which can be explained as follows: The signal strengths of ATI and LAES signals depends on the particle density at the position of laser ionization and the cross section of the multiphoton process. The particle density for the ATI spectra is different for In, Xe and AC because different pickup sources (resistively heated oven for In and gas cell for Xe and AC) are used and operated at slightly different conditions, resulting in different particle flux rates of the effusive beam. Likewise, the (In,Xe,AC)-He_N flux rate varies due to slightly different nozzle conditions ($R_d = 44 \text{ \AA}$ for In and $R_d = 47 \text{ \AA}$ for Xe, AC) and different probabilities to load the droplet with one In/Xe/AC. The most influential factor, however, is likely the difference in multiphoton ionization cross section for the three species in the He environment, which differs presumably from its gas-phase analogue and is unknown. Area-normalization allows us to circumvent these complications and to directly compare yields as function of kinetic energy or for different species, as the area normalized measured spectrum is an approximation for the underlying electron spectrum.

The noise level is not changed due to area normalization because we use semi-logarithmic plots, where multipli-

cation of the spectrum with a normalization factor leads to a vertical shift and does not scale up certain regions (as can be seen by comparison of the corresponding figures here and in the manuscript).

Reviewer: *As the size of the droplets increases, the spectrum is also shifted to higher kinetic energies (Fig. 2). The authors explain this qualitatively and quantitatively with simulations. The simulations show the number of collisions for variable droplet size and variable electron energy. Here it is noticeable that in the time interval of the pulse (25 fs) up to 40 collisions take place. The experimental results are very good, the simulations add value to these results.*

Response: At this point we would like to emphasize our new Monte Carlo 3D LAES simulations, that not only further support our interpretation in terms of LAES as the accelerating process, but also allow for more reliable quantitative statements like the transit time through the He layer and the corresponding number of scattering events. This is expressed in the revised manuscript at various positions, such as:

- Page 10: "The median value for the electron transit time through the liquid He layer, corresponding to an electron ejection ratio of 0.5 in Figure 3a, increases from 11 fs for $R_d = 32 \text{ \AA}$, to 20 fs for $R_d = 76 \text{ \AA}$, and to 164 fs for $R_d = 340 \text{ \AA}$. "
- Page 10: "The probability distributions of laser-assisted scattering events (Figure 3b) give further insight into the droplet size dependency. The mean number of scattering events increases by a factor of 4 from 6 for $R_d = 32 \text{ \AA}$ to 24 for $R_d = 340 \text{ \AA}$. "

Reviewer: *Regarding major claims and novelty: The major claim in the abstract, introduction, and conclusion is that ultrafast electron microscopy (UEM) with LAES gating is, due to this paper, also possible in condensed samples. I find this statement confusing. Good spatial resolution in UEM requires high kinetic energy of the incoming electrons and high angular information content of the outgoing electrons. Morimoto, Kanya and Yamouchi used 1 keV electrons from an external source and modulated those using LAES, and concentrated on the first sideband for retrieving angular and thus structural information. One could extend the simulations given in Fig. 3 to 1 keV and test the cross-sections, and it will most likely work, but this is not the topic of the paper. Also, experiments with high energy electrons in condensed media are already done with a high spatial resolution (Ischenko et al. Chem. Rev. 217, 16, 11066 give a good review), so the question to ask is if under those conditions the combination with LAES would work. The claim that could be made in this manuscript is that the LAES energy-modulation mechanism is observed for the first time in a condensed, superfluid medium and that multiple LAES interactions can take place in He-nanodroplets and high kinetic energy can be achieved by this. This however works in the multiple-scattering regime. That is an important discovery in strong-field physics and could possibly be of interest to a wider community.*

Response: We thank the reviewer for his/her opinion on the applicability of LAES for structural analysis with

Figure 1: Comparison of ATI and LAES spectra, in analogy to Figure 1 of the manuscript. The spectra are plotted without normalization.

electron probes and, in particular, for his/her suggestion about the relevance of our results for the strong-field physics community. We thank the reviewer for his/her opinion. We follow his/her advice in that we do not speculate about the behaviour at 1 keV since we agree that this energy range is not the topic of the manuscript. Consequently, we have removed sentences concerning ultrafast electron microscopy from Abstract, Introduction and Discussion sections, such as:

- "Our results reveal that LAES could significantly increase the temporal resolution of ultrafast electron microscopy, potentially to the attosecond regime."
- "Consequently, spatial separation of pulse shaping and structural probing in ultrafast electron microscopy (UEM) setups leads to broadening of electron pulses during delivery to the sample, limiting the temporal resolution currently to about 80 fs"
- "For the applicability of LAES in UEM,..."
- "In view of UEM, our results suggest that LAES can increase the temporal resolution of electron probes through optical gating with ultrashort laser pulses."

We fully agree that it is an "important discovery in strong-field physics" that the "LAES energy-modulation mechanism is observed for the first time in a condensed, superfluid medium". We feel that we sufficiently point this claim out throughout the revised manuscript:

- "Here we report on the observation of LAES at condensed phase particle densities,"
- "Supported by Monte Carlo 3D LAES and classical scattering simulations, these results provide the first insight into the interplay of LAES energy gain/loss and dissipative electron movement in a liquid."
- "In conclusion, we have demonstrated that LAES can be observed with femtosecond laser pulses in the condensed phase at particle densities of $2 \cdot 10^{22} \text{ cm}^{-3}$."
- "We show that electrons can be accelerated to high kinetic energies through multiple LAES processes and support our interpretations with Monte Carlo 3D LAES simulations."

Reply to Reviewer 2

We thank the Reviewer 2 for a careful and insightful report. Most importantly, we now fully support our initial interpretation, that was questioned by the reviewer, with high-level theory simulations, which could be obtained in this short time by intensifying an existing collaboration with one of the world's experts in the field, Prof. Reika Kanya from Tokyo Metropolitan University. The results of his Monte Carlo 3D simulation on laser-assisted electron scattering are in nearly perfect agreement with our experiments and leave no doubt about the interpretation in terms of multiple LAES processes.

Reviewer: *This work reports the experimental observation of enhanced emission of energetic electrons generated by ATI of dopants embedded in helium nanodroplets. Similar electron spectra have recently been reported for pure and doped helium nanodroplets [Phys. Rev. Lett. 125, 093202, arXiv:2102.08150], and should be mentioned. The dependence of the electron spectra on the helium droplet size seems to indicate a crucial role of the helium shell around the dopant in enhancing electron emission. Classical elastic electron-helium scattering simulations are performed to get some insight into the interaction of emitted electrons with the helium nanodroplet. The authors put their results into the perspective of ultrafast microscopy. While the reported experimental findings are very interesting and definitely deserve to be published, the present discussion suffers from severe shortcomings which have to be remedied before this manuscript should be further be considered for publication in Nature Communications.*

1) *The given interpretation of the experimental data in terms of LAES is based on a classical simulation of electron-helium binary scattering, which informs about the number of elastic collisions as a function of the initial electron energy and the droplet size. However, since the interaction with the laser pulse is not considered, the connection to LAES remains unclear. As long as the simulation does not provide any information about the enhancement of the emission of energetic electrons by the coupling to the laser field, the interpretation in terms of LAES has to be considered as qualitative / speculative. In particular, it remains unclear why energy absorption by the electrons from the light field dominates over emission of photons leading to electron-energy loss.*

Response: We thank the reviewer for his/her doubts concerning the classical simulations; this point has led to the most important increase in quality of the revised manuscript. In order to include the interaction with the laser pulse in the electron-He scattering simulation, Prof. Reika Kanya, one of the world's experts in this field, has performed Monte Carlo 3D LAES simulations within the parameter range of the experiments (laser intensities and He droplet sizes), the results of which are in astonishing agreement with the measured electron spectra. This agreement leaves no doubt about our interpretation in terms of LAES. Additionally, we are now able to make quantitative statements like the electron transit time through the He and the number of scattering events. The classical scattering simulations, however, still are an important contribution to the manuscript

by quantifying the electron energy dissipation within the largest droplets at long time scales after the laser pulse.

Implementation of the Monte Carlo 3D LAES simulations has led to major modifications throughout the manuscript, including Figures 1a, 2c and 3a,b, as well as corresponding descriptions in the Results and Discussion sections. Importantly, however, the overall conclusion in terms of LAES has not changed but is now backed up by high-level LAES simulations.

In the following we list some of the sentences/paragraphs that have been added in the revised manuscript:

- Page 4: "In order to identify the process causing this strong electron acceleration, we simulate the interaction of the In-He_N system with a light pulse under the same conditions as in the experiment. As described in detail in the Methods section, we obtain the initial electron kinetic energy distribution by assuming ADK-type tunnel ionization¹ and simulate subsequent binary LAES events with He atoms by applying the Kroll-Watson theory². Our simulation calculates 3D electron trajectories within a He droplet of radius R_d and applies a Monte Carlo approach for the LAES events. The simulated LAES spectrum (Figure 1a, blue curve) shows the kinetic energy distribution of electrons that leave the droplet within 400 fs, the time-window of the simulation. The good agreement of the simulated spectrum with the experiment strongly indicates that the observed electron acceleration is due to LAES."
- Page 7: "Simulated LAES spectra for different droplet sizes, shown in Figure 2c, also reveal a very pronounced droplet-size dependence of the electron spectrum. As in the experiment, the energy gain continuously increases with droplet size because larger droplets allow for an increased number of LAES events within the duration of the laser pulse. It is important to realize, however, that the droplet sizes specified for the experiment (Figure 2a) are subject to uncertainty because (i) the generation process of the droplets results in a log-normal size distribution and (ii) the probability to load a droplet with an atom/molecule follows a Poissonian distribution³. The fact that the experiment shows a slightly lower energy gain maximum for $R_d = 340 \text{ \AA}$ (Figure 2a), as compared to the simulations (Figure 2c), indicates that the mean droplet radius of the ensemble observed in the experiment is slightly smaller than 340 \AA . Additional deviations might arise from the assumption of the simulations that the dopant is located at the center of the droplet, whereas the experiment might average over a spatial dopant distribution given by a flat holding potential⁴. Additionally, the LAES simulations follow electron trajectories for 400 fs and only consider electrons that have left the droplet within this time window, resulting in a reduction of low-

¹Ammosov, M. V., Delone, N. B. Krainov, V. P. Tunnel ionization of complex atoms and of atomic ions in an alternating electromagnetic field. *Sov. Phys. JETP* 64, 1191 (1986)

²Kroll, N. M. Watson, K. M. Charged-Particle Scattering in the Presence of a Strong Electromagnetic Wave. *Phys. Rev. A* 8, 804–809 (1973)

³Toennies, J. P. Vilesov, A. F. Superfluid Helium Droplets: A Uniquely Cold Nanomatrix for Molecules and Molecular Complexes. *Angewandte Chemie International Edition* 43, 2622–2648 (2004)

⁴Thaler, B. et al. Conservation of Hot Thermal Spin–Orbit Population of 2P Atoms in a Cold Quantum Fluid Environment. *The Journal of Physical Chemistry A* 123, 3977–3984 (2019)

energy signal for the largest droplets because some electrons are inside the droplet after the time window covered by the simulation. Apart from these minor differences, Figure 2 reveals very good agreement of the predicted and observed droplet size dependence of the LAES process.”

- Page 10: ”The mean value for the electron transit time through the liquid He layer, corresponding to an electron ejection ratio of 0.5 in Figure 3a, increases from 11 fs for $R_d = 32 \text{ \AA}$, to 20 fs for $R_d = 76 \text{ \AA}$, and to 164 fs for $R_d = 340 \text{ \AA}$. The ratio of ejected electrons levels off at $\sim 80 \%$ after 100 fs, indicating that $\sim 20 \%$ of the electrons have not left the droplet by the end of the simulated time window of 400 fs. The initially non-zero ejection probability for the smallest droplets reflects the fact that the simulated laser pulse is centered around at time-zero.

The probability distributions of laser-assisted scattering events (Figure 3b) give further inside into the droplet size dependency. The mean number of scattering events increases by a factor of 4 from 6 for $R_d = 32 \text{ \AA}$ to 24 for $R_d = 340 \text{ \AA}$. ”

Additionally, we now mention the requested reference in the introduction: ”Such advantages recently enabled the application of above threshold ionization (ATI) to He droplets³².”

Reviewer: 2) *The validity of the scattering model is very limited; besides the mentioned neglect of the interaction of the electron with its parent ion and with the light field, inelastic scattering is disregarded, although cross sections are about 0.1 \AA^2 At. Data Nucl. Data Tables 94, 603 (2008)]. inelastic scattering of a photoelectron upon a neighboring helium was observed [J. Chem. Phys. 150, 044304 (2019)]. More importantly, it is well known that electron-helium collisions at low energies 1eV are crucially influenced by the collective response of the helium superfluid causing the electron to be trapped in a bubble [Phys. Rev. Lett. 75, 4079]; attachment of electrons to helium droplets at energies as high as 10 eV was observed [J. Chem. Phys. 108, 9327 (1998)]. Electrons were found to recombine with impact-excited He atoms and molecules [J. Phys. Chem. Lett. 2014, 5, 2444, J. Phys. Chem. A 2014, 118, 66426647]. Moreover, the assumption made by the authors that dopants are located at the center of the droplets may not be justified, at least for the larger droplet they used. The binding potential of the dopants to the droplets is known to be rather flat which leads to some delocalization of the dopants [Mol. Phys, 1999, 97, 645 666].*

Response: We thank the reviewer for raising these important points and agree that they should be included in future developments of the simulations. On the other hand, the very good agreement of our LAES simulations in the current form with the experiment makes us feel confident that they are sufficient to confirm our interpretation in terms of LAES, which is the aim of this manuscript. Nevertheless we would like to comment on each of these points in the following:

- Concerning the ”neglect of the interaction of the electron with its parent ion”:

We agree with the reviewer. Indeed, the charge of the remaining parent ions, i.e. the ions’ Coulomb fields, should have some influence on the electrons’ trajectories and therewith on the energy absorption from the light

field. The neglect of this influence, commonly made in the interpretation of experiments that investigate the interaction of atoms (or molecules) with strong laser fields, known as the strong-field approximation (SFA)⁵],⁶, fails for very low electron-energies⁷, but leads to an at least qualitatively acceptable agreement with experiments for higher electron energies. In our experiments we analyze electrons with energies in excess of 10 eV. For this electron energy range it is considered well justified that the Coulomb potential of the parent ion does not significantly contribute to the electron energy gain/loss mechanism.

b. Concerning *"the interaction of the electron with the light field"*:

Interaction with the light field is now included (see above).

c. Concerning *"inelastic scattering"*:

We agree that in view of the work by Marcel Mudrich and coworkers⁸ inelastic scattering should play a role and one would expect a corresponding signature in the LAES spectra (Figure 1 of the main manuscript), most probably a change in slope. However, the fact that no such signature is apparent around 20 eV in the spectra, indicates that inelastic scattering processes seems to play a minor role. This is supported by the much larger cross section for elastic scattering at 20 eV of 2.7 \AA^2 (see Figure 1c in the main manuscript and Brunger et al.⁹), which exceeds the inelastic cross section of 0.1 \AA^2 by a factor of 27.

Further supported can be found the paper of Henne et al.¹⁰ (which was suggested by the Reviewer), who write in their appendix: *"... Assuming the cross sections for elastic scattering and electronic excitation in the droplet to be equal to the gas phase values, one finds that on average approximately 100 elastic collisions occur before one inelastic collision with 3S excitation can take place at $E \sim 21 \text{ eV}$..."* For 42 eV and 63 eV electrons this factor increases to 150, and 250 respectively.

We mention this on page 5 of the manuscript: *"Furthermore, the absence of a kink in the yield at or above 20 eV, the energy threshold of electronic He excitations¹¹, shows that inelastic interactions are insignificant, which is in agreement with the much lower cross section for inelastic as compared to elastic interaction¹²."*

⁵F. Faisal, "Multiple absorption of laser photons by atoms," J. Phys. B At. Mol. 6, L89 (1973); [<http://dx.doi.org/10.1088/0022-3700/6/4/011>]

⁶H. Reiss, "Effect of an intense electromagnetic field on a weakly bound system," Phys. Rev. A 22, 1786–1813 (1980); [<http://dx.doi.org/10.1103/PhysRevA.22.1786>].

⁷X. Xie, S. Roither, S. Gräfe, D. Kartashov, E. Persson, C. Lemell, L. Zhang, M. S. Schöffler, A. Baltuška, J. Burgdörfer, and M. Kitzler, "Probing the influence of the Coulomb field on atomic ionization by sculpted two-color laser fields," New J. Phys. 15, 043050 (2013); [<http://dx.doi.org/10.1088/1367-2630/15/4/043050>].

⁸M. Shcherbinin et al., "Inelastic scattering of photoelectrons from He nanodroplets," J. Chem. Phys. 150, 044304 (2019), <https://aip.scitation.org/doi/10.1063/1.5074130>

⁹M. J. Brunger, S. J. Buckman, L. J. Allen, I. E. McCarthy, and K. Ratnavelu, "Elastic electron scattering from helium: absolute experimental cross sections, theory and derived interaction potentials," Journal of Physics B: Atomic, Molecular and Optical Physics 25, 1823 (1992), <https://doi.org/10.1088%2F0953-4075%2F25%2F8%2F016>

¹⁰U. Henne, J.P. Toennis, "Electron capture by large helium droplets", Journal of Chemical Physics 108, 9327 (1998)

¹¹Kramida, A., Yu. Ralchenko, Reader, J. and NIST ASD Team. NIST Atomic Spectra Database (ver. 5.8), [Online]. Available:<https://physics.nist.gov/asd>[2020, November 5]. National Institute of Standards and Technology, Gaithersburg, MD. 2020

¹²M. J. Brunger, S. J. Buckman, L. J. Allen, I. E. McCarthy, and K. Ratnavelu, "Elastic electron scattering from helium: absolute experimental cross sections, theory and derived interaction potentials," Journal of Physics B: Atomic, Molecular and Optical Physics 25, 1823 (1992), <https://doi.org/10.1088%2F0953-4075%2F25%2F8%2F016>

d. Concerning *"the collective response of the helium superfluid"*:

We agree that interaction of electrons with He droplet as a whole and its collective modes are to be expected. These interactions are primarily in the few-eV range, as documented by the references mentioned by the reviewer. Since there are not signatures for distortion in the low-energy range of our spectra, we assume that the influence due to collective excitations is minor. Importantly, the main conclusions in our manuscript are drawn from electrons with energies in excess of 10 eV, which are not biased in terms of collective He phenomena.

e. Concerning *"the center location of the dopant"*:

We agree that the dopant distribution, as given by the holding (binding) potential within the droplet, might influence the simulated LAES spectra. This will be included in future simulations. In the current manuscript, we actually find a small deviation of experiment and theory for the largest droplets (Figures 2a and 2c of the main manuscript), which might be due to a spatial distribution of the dopant, as now mentioned on page 8 of the manuscript:

"Additional deviations might arise from the assumption of the simulations that the dopant is located at the center of the droplet, whereas the experiment might average over a spatial dopant distribution given by a flat holding potential."

Reviewer: 3) *The role of multiple dopants being present in one droplet needs to be carefully discussed. Due to the broad size distribution of the He droplets, even at moderate average droplet sizes there are contributions to the signals from much larger droplets which have large pickup cross sections. The authors should specify how they made sure that the droplets were doped by single atoms / molecules at all shown droplet sizes.*

Response: We have determined the pickup conditions with a quadrupole mass spectrometer. In addition we have carefully checked for influence of multi-particle doping, as we were concerned about blurring of the ATI/LAES contrast due to molecular dynamics (we suppose that this is also the concern of the reviewer). For In-He_N, for example, LAES spectra show surprisingly little contrast decrease for increased pickup conditions (Figure 2): An increase of the pickup conditions from single In atom pickup (43 A heating current of the pickup source, as used for the $R_d = 76 \text{ \AA}$ measurements in the main manuscript) to above the maximum for pickup of two In atoms (49 A) only led to a small decrease of the oscillation contrast. We assume that our laser pulses are sufficiently short (25 fs) to avoid blurring effects from nuclear dynamics of, e.g., In₂. Similarly, for acetone molecules (Figure 1c in the main manuscript) intramolecular dynamics seem not to play a major role, which might be an issue for longer laser pulses.

Figure 2: LAES spectra of In-He_N for $R_d = 76 \text{ \AA}$ droplets under different doping conditions.

Reviewer: 4) An important motivation for the present work given by the authors is the relevance of their findings for ultrafast electron microscopy. As they don't show any angle-resolved data, this claims remains unfounded and questionable.

Response: We agree and have, in consensus with Reviewer 1 (comment 3), removed the outlook on the application of LAES for ultrafast electron microscopy.

Reviewer: Minor issues:

p. 2: Sentence starting with “Time-domain shaping of electron pulses. . .”; citing references from 1923 and 1953 in this context seems awkward.

Response: We agree and now provide the reader with a more modern perspective for these two effects (Smith-Purcell effect and Compton scattering) by adding two references to the historic papers: (i) A review article on electron microscopy¹³ that discusses the generation of radiation through the Smith-Purcell effect in electron microscopes as the beam passes by periodic structures. (ii) A textbook on Compton scattering¹⁴, that provides an overview on applications of the Compton effect, such as testing electron densities or spin distributions.

Reviewer: p. 2: Sentence starting with “Consequently, spatial separation of pulse shaping and structural

¹³García de Abajo, F. J. Optical excitations in electron microscopy.Reviews of Modern Physics 82, 209–275 (2010).

¹⁴Cooper, M.et al. X-Ray Compton Scatteringisbn: 9780198501688 (OUP Oxford, 2004).

probing...” is unclear; what is meant by “spatial separation of pulse shaping...”?

Response: This sentence has been removed. What we meant, however, was that in UEM setups the formation/shaping of electron pulses is spatially separated from the interaction region with the sample, where structural probing occurs. Within this distance the electron pulse duration increases due to velocity dispersion and Coulombic repulsion.

Reviewer: *p. 10: Sentence ending with “...because it is influenced only by the structure of the immediate environment of the dopant, the solvation shell.” contradicts previous findings of droplet-size dependent electron spectra [J. Phys. Chem. A, 112, 2008, 9359].*

Response: We assume the reviewer is referring to J. Phys. Chem. A, 112, 2008, **9356** (instead of page 9359). We thank the reviewer for mentioning this potential misunderstanding. The droplet size dependence of the photoelectrons in this paper reveals significant energy loss of electrons on their way out of large droplets, leading to a structure-less photoelectron band below 5 eV. The mentioned sentence, in contrast, refers to energy gain of 50 to 100 eV. To clarify this we have extended the sentence in question on page 13 of the manuscript:

”This behavior observed for strong-field ionization is in contrast to weak-field ionization inside He droplets, where the photoelectron spectrum is **either** droplet-size independent because it is influenced only by the structure of the immediate environment of the dopant, the solvation shell¹⁵, **or develops a low-energy band revealing significant energy loss of electrons in larger droplets**¹⁶.”

¹⁵Thaler, B., Heim, P., Treiber, L. Koch, M., Ultrafast photoinduced dynamics of single atoms solvated inside helium nanodroplets. J. Chem. Phys. 152, 014307 (2020)

¹⁶Wang, C. C. et al. Photoelectron Imaging of Helium Droplets Doped with Xe and Kr Atoms. The Journal of Physical Chemistry A 112, 9356–9365 (2008).

Reply to Reviewer 3

We thank Reviewer 3 for his/her report, in particular for mentioning the laser-assisted photoelectric effect, which is now discussed in the manuscript and provides the reader with a broader context on electron-light interaction.

Reviewer: *The work presented by Treiber et al. reports on strong field ionization of atoms and molecules surrounded by a helium layer. The interaction of the generated electrons with the helium in the presence of the strong light field leads to the generation of sidebands in the photoelectron spectrum separated by the photon energy. The authors refer to this process as Laser-assisted electron scattering (LAES) and claim in the abstract that this has only be observed in the gas phase. However, the process is not fundamentally different from the Laser-Assisted Photoelectric Effect (LAPE), where usually a weak UV laser ionizes species in the presence of a strong laser pulse. These types of experiments have been performed in gas, liquid and solid phase. The experiments as described the authors are thus not the first of this kind in the liquid phase as they claim. Their study does provide some additional insight into the process, i.e. to what extent the size of the helium contributes to the shift of photoelectron energy. In my view, these results are however not of broad interest to warrant publication in Nature Communication.*

Response: We agree that the LAPE process should be mentioned in the manuscript. In order to put LAPE in the right context for the reader, we consider it important to point out that the energy modulation in LAPE occurs during the ionization event, which is the basis for all the remarkable applications, like clocking the emission delay of conduction-band and core electrons¹⁷. LAES, in contrast is sensitive to the interaction of electrons with neutral particles, in a bulk liquid also in multiple events, detached from the ionization process. To cover this, we have added/changed the following sentences in the Introduction section:

”Other strong-field phenomena like high-order harmonic generation¹⁸ have been extended from the gas phase to solid-state systems, providing insight into the attosecond electron dynamics and non-equilibrium situations in band structures. Also, the laser-assisted photoelectric effect was demonstrated from the surface of a solid¹⁹, allowing to map the electron emission process with attosecond resolution²⁰. LAES, in contrast, where an electron probes the structure of neutrals far away from its origin, has evaded observation in the condensed phase so far, so that its potential for advancing time-resolved structural probing at high particle densities remains unexplored.

¹⁷Cavalieri, A. L. et al. Attosecond spectroscopy in condensed matter. Nature 449, 1029–1032 (2007).

¹⁸Ghimire, S. et al. Observation of high-order harmonic generation in a bulk crystal. Nat Phys 7, 138–141 (2010).

¹⁹Miaja-Avila, L. et al. Laser-Assisted Photoelectric Effect from Surfaces. Physical Review Letters 97, 113604 (2006)

²⁰Cavalieri, A. L. et al. Attosecond spectroscopy in condensed matter. Nature 449, 1029–1032 (2007).

Reviewer: *In general, the authors would benefit from a more exhaustive study of the literature.*

Response: In the revised version of the manuscript we have added references for the following fields:

- Laser-assisted photoelectric effect: Ref. 28 [Miaja-Avila, L. et al. Laser-Assisted Photoelectric Effect from Surfaces. *Physical Review Letters* 97, 113604 (2006).]
Ref. 29 [Cavalieri, A. L. et al. Attosecond spectroscopy in condensed matter. *Nature* 449, 1029–1032 (2007).]
- Time-resolved electron diffraction: Ref. 9 [Ischenko, A. A., Weber, P. M. Miller, R. D. Capturing chemistry in action with electrons: realization of atomically resolved reaction dynamics. *Chemical reviews* 117, 11066–11124 (2017).]
- Above-threshold ionization: Ref. 34 [Kelbg, M. et al. Temporal Development of a Laser-Induced Helium Nanoplasma Measured through Auger Emission and Above-Threshold Ionization. *Physical Review Letters* 125, 093202 (2020).]
- Photoelectron spectroscopy in He droplets: Ref. 41 [Wang, C. C. et al. Photoelectron Imaging of Helium Droplets Doped with Xe and Kr Atoms. *The Journal of Physical Chemistry A* 112, 9356–9365 (2008).]
- Electron microscopy: Ref. 11 [García de Abajo, F. J. Optical excitations in electron microscopy. *Reviews of Modern Physics* 82, 209–275 (2010).]

These add to the fields referenced in the initial version:

- Ultrafast electron microscopy: Ref. 2, 3, 5, 8
- Ultrafast low-energy electron diffraction: Ref. 6
- Photon-induced near-field electron microscopy (PINEM): Ref. 4, 7
- Free electron lasers: Ref. 15
- High-order harmonic generation: Ref. 16, 27
- High harmonic spectroscopy: Ref. 17
- Laser-induced electron diffraction: Ref. 18
- Electron-pulse emission from metal tips: Ref. 19, 20
- Generation of electron pulse trains: Ref. 21, 22
- Laser-assisted electron diffraction: Ref. 23-26, 35, 36, 47
- He droplets: Ref. 30, 31, 33, 39, 41, 42, 43, 44, 46

Reviewer: *As a motivation for their work they refer to ultrafast electron microscopy, stating that the best resolution obtained is 60 fs. However, Baum and coworkers in Munich recently recorded electron images and diffraction patterns with attosecond resolution. It remains questionable if the authors will be able to achieve such resolution based on LAES, even more so since they do not provide any indication how they would perform such experiments.*

Response: We thank the reviewer for noting that electron pulse trains can actually reach a much shorter duration at the refocusing distances²¹ than isolated electron pulses²², which, on the other side, allow for classical pump-probe delay-scan studies. However, we follow the advice of Reviewers 1 and 2 by bringing to the foreground that "the LAES energy-modulation mechanism is observed for the first time in a condensed, superfluid medium", so that a discussion of advantages and disadvantages of isolated electron pulses and electron pulse trains becomes obsolete in the revised version of the manuscript.

Reviewer: *As for the reported experiments, there is one issue the authors should address. They mention that in their simulations on the electron motion thorough helium they neglect the acceleration/deceleration due to LAES, and the momentum transfer. However, they do not mention anything whether the charge of the remaining ion is taken into account and what effect this might have on the calculations.*

Response: Acceleration and deceleration due to LAES is now included in our simulations, see Reviewer 2, comment 1. Concerning the neglect of the ions, we agree with the reviewer. Indeed, the charge of the remaining parent ions, i.e. the ions' Coulomb fields, should have some influence on the electrons' trajectories and therewith on the energy absorption from the light field. The neglect of this influence, commonly made in the interpretation of experiments that investigate the interaction of atoms (or molecules) with strong laser fields, known as the strong-field approximation (SFA)^{23 24}, fails for very low electron-energies²⁵, but leads to an at least qualitatively acceptable agreement with experiments for higher electron energies. In our experiments we analyze electrons with energies in excess of 10 eV. For this electron energy range it is considered well justified that the Coulomb potential of the parent ion does not significantly contribute to the electron energy gain/loss mechanism.

We mention this issue on page 14 of the revised manuscript as one of the shortcomings of the simulation:

"This discrepancy points towards shortcomings of the simulation that are currently neglected: excitation of collective droplet modes^{28,29}, transit-time increase due to Coulomb interaction between the ion core and the

²¹Morimoto, Y. Baum, P. Diffraction and microscopy with attosecond electron pulse trains. *Nature Physics* 14, 252–256 (2018).

²²Ryabov, A. Baum, P. Electron microscopy of electromagnetic waveforms. *Science* 353, 374–377 (2016).

²³F. Faisal, "Multiple absorption of laser photons by atoms," *J. Phys. B At. Mol.* 6, L89 (1973); [<http://dx.doi.org/10.1088/0022-3700/6/4/011>].

²⁴H. Reiss, "Effect of an intense electromagnetic field on a weakly bound system," *Phys. Rev. A* 22, 1786–1813 (1980); [<http://dx.doi.org/10.1103/PhysRevA.22.1786>].

²⁵X. Xie, S. Roither, S. Gräfe, D. Kartashov, E. Persson, C. Lemell, L. Zhang, M. S. Schöffler, A. Baltuška, J. Burgdörfer, and M. Kitzler, "Probing the influence of the Coulomb field on atomic ionization by sculpted two-color laser fields," *New J. Phys.* 15, 043050 (2013); [<http://dx.doi.org/10.1088/1367-2630/15/4/043050>].

electron, or additional blurring of the LAES peaks due to sequential energy-gain–energy-loss processes with varying U_P .”

Reviewers' Comments:

Reviewer #1:

Remarks to the Author:

I would like to thank the authors for seriously considering my suggestions in the current version of the manuscript.

In agreement with some of the other reviewers, the motivation regarding ultrafast electron microscopy, has been removed.

Also my concerns regarding the experimental data as well as the simulations have been sufficiently clarified.

Reviewer #2:

Remarks to the Author:

In the revised version of the manuscript, the authors have indeed rebutted my main concern – the lacking correspondence of the scattering simulations with the experimental data. The new simulations now impressively confirm the interpretation of the electron spectra in terms of LAES. The other points I had raised are also addressed more or less convincingly. Only the question about multiple doping still remains somewhat vague. The authors just state “We have determined the pickup conditions with a quadrupole mass spectrometer.” How do the authors infer the accurate number of dopants per droplet from the fragment-ion mass spectra measured with the QMS? The doping conditions are specified in terms of heating current, which is not a meaningful quantity as it depends on the specific construction of the oven; doping conditions should be specified in terms of the dopant partial pressure in the scattering cell, knowing the spatial dimensions of that cell.

Furthermore, I request a (fair) discussion of the relevance of these findings for other nanostructures such as heavier rare-gas clusters, molecular clusters, other (metallic, semiconductor, etc.) nanoparticles. Is the observed effect specific to helium, possibly related to the extremely high ionization energy which allows to apply relatively high laser intensities before igniting the droplets? Is the simple electronic structure (no inner electron shells) of helium particularly advantageous? Does superfluidity play any role?

Given the nice experimental results for an interesting system – doped superfluid helium nanodroplets – and the convincing interpretation supported by simulations, I would recommend publication the above points are solved. This is the first demonstration of LAES for unsupported core-shell nanostructures and one may envision implications for and applications in the broader fields of strong-field physics and ultrafast spectroscopy.

Marcel Mudrich

Reviewer #3:

Remarks to the Author:

The motivation of the work as reported in the original manuscript, LAES being a tool for ultrafast microscopy is almost completely removed in the revised manuscript. The work is now presented in the context of strong field physics, which in my view limits its potential interest to the broad audience of Nature Communications.

On the other hand, the manuscript has been substantially improved by including simulations taking explicitly into account LAES. The excellent agreement between the experiential data and the simulations provide convincing evidence that laser-assisted electron scattering is the at the origin of the observed kinetic energy distributions. This work will therefore certainly be of interest to the broader community of strong field physics. If the authors would be able to put this work in a

somewhat broader context it would certainly fulfill the criteria for publication in Nature Communications.

Reply to Reviewer 1

***Reviewer:** I would like to thank the authors for seriously considering my suggestions in the current version of the manuscript. In agreement with some of the other reviewers, the motivation regarding ultrafast electron microscopy, has been removed. Also my concerns regarding the experimental data as well as the simulations have been sufficiently clarified.*

Response: We thank Reviewer 1 for the positive report.

Reply to Reviewer 2

Reviewer: *In the revised version of the manuscript, the authors have indeed rebutted my main concern – the lacking correspondence of the scattering simulations with the experimental data. The new simulations now impressively confirm the interpretation of the electron spectra in terms of LAES. The other points I had raised are also addressed more or less convincingly. Only the question about multiple doping still remains somewhat vague. The authors just state “We have determined the pickup conditions with a quadrupole mass spectrometer.” How do the authors infer the accurate number of dopants per droplet from the fragment-ion mass spectra measured with the QMS? The doping conditions are specified in terms of heating current, which is not a meaningful quantity as it depends on the specific construction of the oven; doping conditions should be specified in terms of the dopant partial pressure in the scattering cell, knowing the spatial dimensions of that cell.*

Response: In order to determine the pickup conditions, we recorded the monomer, dimer, trimer ion signals (e.g., In^+ , In_2^+ , In_3^+ , ...) with the QMS as function of the current of the resistively heated pickup cell. We determine the optimal heating current for the experiment based on the onset of the monomer and dimer ion signals, keeping in mind the possibility of fragmentation and the corresponding bias of monomer/multimer signals towards higher heating currents. We specify the doping condition in terms of heating current because the temperature measurement of our pickup cell is unfortunately very inaccurate. In combination with the power-law dependency of the vapor pressure this would result in an unacceptably high uncertainty of the calculated particle density within the pickup cell.

We point out that even for multiply doping conditions we face a different situation compared to the recently published study by Michiels et al. (arXiv:2105.01918), where XUV photoexcitation of pure droplets promotes several He atoms within a single droplet to a photoexcited state, resulting in increased photoelectron kinetic energies. In our case of doped droplets, in contrast, the electrons originate from a single particle (the dopant) within the droplet. Even in case of multiple dopant pickup, van der Waals interaction leads to aggregation of the dopants to a single cluster; multicenter aggregation is expected to set in at much higher doping rates (see Phys. Rev. Lett. 106 (2011) 233401). Therefore, the only influence of multiple doping on our LAES spectra would be due to a different initial electron spectrum of monomers compared to dimers/trimers, which we have carefully tested: The shape of the LAES spectra does not change within a wide range of doping conditions (see Figure 2 in our first reply to Reviewer 2). These results clearly show that our LAES spectra do not depend on the number of dopant atoms/molecules.

We have added the the following paragraph to the Methods section in order to discuss this important topic: ”We further monitor the pickup conditions by recording the monomer, dimer, and trimer ion signals (e.g., In^+ , In_2^+ , In_3^+) with a quadrupole mass spectrometer as function of the current of the resistively heated pickup cell. When changing the droplet size we ensure constant pickup conditions by adapting the particle density

within the pickup region accordingly. Since loading the He droplets is a statistical process, we have carefully checked if the presence of multiple dopants within one droplet influences the LAES spectra. In the range of, on average, one to three In atoms per droplet we find no significant change of the spectra which can be rationalized by the following two aspects: First, multimer formation due to van der Waal interaction between individual dopants leads to single ionization centers even in multiply doped droplets and, second, the initial photoelectron spectrum of these multimers is size-independent and similar to that of the monomer.”

Reviewer: *Furthermore, I request a (fair) discussion of the relevance of these findings for other nanostructures such as heavier rare-gas clusters, molecular clusters, other (metallic, semiconductor, etc.) nanoparticles. Is the observed effect specific to helium, possibly related to the extremely high ionization energy which allows to apply relatively high laser intensities before igniting the droplets? Is the simple electronic structure (no inner electron shells) of helium particularly advantageous? Does superfluidity play any role?*

Response: These aspects are now addressed in the Discussion section:

”The significant acceleration of electrons and its dominance of over energy dissipation within liquid He is likely related to the outstanding properties of this rare-gas element: The application of high light-field intensities resulting in strong LAES energy gain is enabled by the exceptionally high ionization energy of He and the high excitation energy prevents inelastic electron collisions up to 20 eV. In heavier rare-gas clusters, LAES can be expected, too, albeit less pronounced. The contribution of the droplets’ superfluid character to the observed energy modulation cannot be deduced from the present results and remains to be investigated, for example with non-superfluid ^3He droplets or mixed $^3\text{He}/^4\text{He}$ droplets¹. It will also be important to investigate the ratio of light-induced energy gain and energy dissipation in other materials, like molecular, metal or semiconductor clusters, the creation of which is facilitated by the very flexible opportunities provided by the He droplet approach for the creation of tailor-made bi-material core-shell nanostructures within the droplet^{2 3}. Photoionization of the core will allow to observe LAES-acceleration and energy dissipation within the shell material.”

Reviewer: *Given the nice experimental results for an interesting system – doped superfluid helium nanodroplets – and the convincing interpretation supported by simulations, I would recommend publication the above points are solved. This is the first demonstration of LAES for unsupported core-shell nanostructures and one may envision implications for and applications in the broader fields of strong-field physics and ultrafast spectroscopy.*
Marcel Mudrich

¹Grebenev, S., Toennies, J. P. Vilesov, A. F. Superfluidity within a small helium-4 cluster: The microscopic Andronikashvili experiment. *Science* 279, 2083–2086 (1998)

²Haberfehlner, G. et al. Formation of bimetallic clusters in superfluid helium nanodroplets analysed by atomic resolution electron tomography. *Nat. Commun.* 6, 8779 (2015)

³Messner, R., Ernst, W. E. Lackner, F. Shell-Isolated Au Nanoparticles Functionalized with Rhodamine B Fluorophores in Helium Nanodroplets. *The Journal of Physical Chemistry Letters* 12, 145–150 (2020)

Response: We thank Reviewer 2 for the very positive conclusion on our results and interpretation, as well as the recommendation for publication.

Reply to Reviewer 3

Reviewer: *The motivation of the work as reported in the original manuscript, LAES being a tool for ultrafast microscopy is almost completely removed in the revised manuscript. The work is now presented in the context of strong field physics, which in my view limits its potential interest to the broad audience of Nature Communications.*

On the other hand, the manuscript has been substantially improved by including simulations taking explicitly into account LAES. The excellent agreement between the experiential data and the simulations provide convincing evidence that laser-assisted electron scattering is the at the origin of the observed kinetic energy distributions. This work will therefore certainly be of interest to the broader community of strong field physics. If the authors would be able to put this work in a somewhat broader context it would certainly fulfill the criteria for publication in Nature Communications.

Response: We thank Reviewer 3 for her/his positive comments on the quality of our results and her/his assessment of the relevance to the strong field community. To put our work in a somewhat broader context, as suggested, we have added the following paragraph to the Discussion section:

”Our results indicate that LAES is a strong-field light–matter interaction process that is, in analogy to high harmonic generation⁴, capable of spatio-temporal analysis of solids. It can be anticipated that LAES has the potential to significantly increase the temporal resolution of electron probes through optical gating, thereby merging temporal selection via velocity modulation of electrons with ultrashort laser pulses (as demonstrated here), and structural analysis that can be extracted from the electron angular distributions^{5 6}.”

⁴Ghimire, S. et al. Observation of high-order harmonic generation in a bulk crystal. Nat Phys 7, 138–141 (2010)

⁵Morimoto, Y., Kanya, R. Yamanouchi, K. Laser-assisted electron diffraction for femtosecond molecular imaging. The Journal of Chemical Physics 140, 064201 (2014)

⁶Kanya, R. Yamanouchi, K. Femtosecond Laser-Assisted Electron Scattering for Ultrafast Dynamics of Atoms and Molecules. Atoms 7, 85 (2019)

Reviewers' Comments:

Reviewer #2:

Remarks to the Author:

The authors have responded to all my questions satisfactorily. I recommend publication of the manuscript without further modifications.

Marcel Mudrich

Reply to Reviewer 2

***Reviewer:** The authors have responded to all my questions satisfactorily. I recommend publication of the manuscript without further modifications. Marcel Mudrich*

Response: We thank Reviewer 2 for the positive report.